



# Estuarine morphodynamics and development modified by floodplain formation

Maarten G. Kleinhans[1], Lonneke Roelofs[1], Steven A.H. Weisscher[1], Ivar R. Lokhorst[1,2], and Lisanne Braat[1,3]

[1]Department of Physical Geography, Utrecht University, Princetonlaan 8A, 3584 CB, Utrecht, The Netherlands
[2]Nelen & Schuurmans, Zakkendragershof 34-44, 3511 AE, Utrecht, The Netherlands
[3]European Space Agency (ESA), European Space Research and Technology Centre (ESTEC), Keplerlaan 1, 2201 AZ Noordwijk, The Netherlands

**Correspondence:** Maarten Kleinhans (m.g.kleinhans@uu.nl)

**Abstract.** Rivers and estuaries are flanked by floodplains built by mud and vegetation. Floodplains affect channel dynamics and the overall system's pattern through apparent cohesion in the channel banks and through filling of accommodation space and hydraulic resistance. For rivers, effects of mud, vegetation and the combination are thought to stabilise the banks and narrow the channel. However, the thinness of mudflats and salt marsh in estuaries compared to channel depth raises questions

about the effects of floodplain as constraints on estuary dimensions. To test these effects, we created three estuaries in a tidal flume: one with mud, one with recruitment events of two live vegetation species and a control with neither. Both mud and vegetation reduced channel migration and bank erosion and stabilised channels and bars. Effects of vegetation include local flow velocity reduction and concentration of flow into the channels, while flow velocities remained higher over mudflats. On the other hand, the lower reach of the muddy estuary showed more reduced channel migration than the vegetated estuary. The

main system-wide effect of mudflats and salt marsh is to reduce the tidal prism over time from upstream to downstream. The landward reach of the estuary narrows and fills progressively, particularly for the muddy estuary, which effectively shortens the tidally influenced reach and also reduces the tidal energy in the seaward reach and mouth area.

## 1 Introduction

The size and shape of natural estuaries are potentially modified by life, as they are often flanked by mudflats and salt marsh

(Whitfield et al., 2012). In analogy with rivers, where floodplains formed by vegetation and mud deposition affect the channel dimensions and channel-bar pattern (see for review Kleinhans, 2010), it is plausible that floodplains in estuaries have similar effects. However, there is in principle no limit to the available flow discharge from the sea, unlike in rivers where long-term discharge frequency and magnitude are determined by hinterland characteristics and climate. This raises the question how vegetation and mud sedimentation affect the channel pattern in estuaries as far as such estuaries are not laterally constrained by

valley walls. The objective is to understand the different mechanisms by which vegetation or mud affect local channel dynamics and system-scale effects on the channel-bar pattern. Two alternative hypotheses for effects of floodplains on the dimensions of estuaries are proposed here from observations and mechanisms in rivers.



The first hypothesis relates to the conservation of tidal energy in the landward direction. In some systems, floodplain formation along estuaries may have led to relatively deep channels allowing further landward tidal penetration than in shallow

estuaries. Estuaries that formed as part of tidally-dominated deltas such as the Mekong, Mahakam, Yangtze and Rhine developed narrow, deep channels surrounded by high floodplains (Tamura et al., 2012; Wang et al., 2015; de Haas et al., 2019), as did some ingressive estuaries (van der Spek, 1997). The channel depth and convergence possibly cause the tides to propagate far landwards with backwater effects up to the delta apex.

The alternative, opposing hypothesis relates to landward tidal energy reduction. The reduction of intertidal area by accreting

floodplains with vegetation reduces the tidal energy, especially on the fluvial-tidal transition, and pushes the tidal limit in the seaward direction. Indeed, on longer timescales, floodplain formation along estuaries was concurrent with a reduction in tidal discharge, landward tidal penetration and overall filling that reduced tidal system extent and depth over some portion of the Holocene (Woodroffe et al., 2016; de Haas et al., 2018; de Haas et al., 2019). Modelling showed that mudflat sedimentation alone is a sufficient condition for net infilling (Braat et al., 2017; Boechat Albernaz et al., 2020) while absence of fine sediment

is known to lead to drowning (van der Wegen, 2013) or keeps estuaries unfilled (Dalrymple and Choi, 2007).

The expansion or filling of estuaries not only depends on mudflat and salt marsh formation but also on the morphodynamic interactions between river discharge and tides and the channels and bars morphology. While significant river discharge causes net seaward sediment transport (Savenije, 2015; Dronkers, 2017), sea-level rise, subsidence and shipping fairway deepening cause flood-dominance that promotes sediment import (Friedrichs and Aubrey, 1988; Brown and Davies, 2010; Wang et al.,

2015). However, the reconstruction of Holocene coastal plains shows that salt marsh and mangrove development likely amplified vertical accretion (Woodroffe et al., 2016; de Haas et al., 2018) in agreement with modelling (Kirwan et al., 2016; Lokhorst et al., 2018; Boechat Albernaz et al., 2020; Brückner et al., 2020). The underlying cause for the reducing planform estuary dimensions may be the reduction of tidal prism (Friedrichs and Aubrey, 1988; Robins and Davies, 2010; Brown and Davies, 2010) by sediment filling intertidal storage area and vegetation causing hydraulic resistance and amplifying vertical accretion.

In other words, there is a direct effect of mud and vegetation by filling and an indirect effect on the tidal dynamics that may determine net sediment import or export.

Intertidal mudflats and supratidal marshes, riparian forest, mires and swamps in the estuarine landscape are important ecosystems along the river continuum and in the coastal zone (e.g. Ysebaert et al., 2016; Woodroffe et al., 2016; de Haas et al., 2018; FitzGerald and Hughes, 2019). Vegetation protects dikes against wave attack and enhanced sedimentation may eventually pro-

vide fertile agriculture soils and may protect urban areas against flooding (Bouma et al., 2014). The channelised parts function as shipping fairways to provide access to the world's major ports (Wang et al., 2015). Understanding floodplain-forming processes is important in view of the projected sea-level rise acceleration, ongoing attempts to build a natural elevated buffer around estuaries and the creation of intertidal basins along estuaries to reduce flood levels. Whether this is a viable pathway depends on the effects of such floodplain dynamics on the large-scale estuarine flow and morphodynamics (e.g. Stark et al.,

2017; Leuven et al., 2019).

Effects of mud or vegetation in isolation cannot be inferred well from observations in nature. On the other hand, numerical models are useful tools to study effects of mud and vegetation in isolation and combination in both rivers and estuaries (e.g.

Kleinhans et al., 2018; Brückner et al., 2021). However, such models are also sensitive to choices of a host of parameterizations of flow resistance, sediment transport (Baar et al., 2019), mud characteristics (Braat et al., 2017) and vegetation characteristics

(Brückner et al., 2020). Landscape experiments, as presented here, may complement numerical modelling by real physical and biological processes at the cost of some known scale effects (Kleinhans et al., 2015b). Recent advances in tidal experiments allow for the formation of estuarine systems with similarities in planform pattern of channel and bar morphology and mudflats and the long-term development as well as response to dredging and sediment disposal (Leuven et al., 2018c; Braat et al., 2019; van Dijk et al., 2021). These experiments focussed on partially filled, multi-channel estuaries with mid-channel bars that

have also been studied in data (Leuven et al., 2018b, 2019) and models (Braat et al., 2017; Baar et al., 2019; Brückner et al., 2020, 2021; van Dijk et al., 2021). Here, we also focus on this type of estuary (examples in Fig. 1).

## 2 Methods

Three estuaries were formed in experiments in the Metronome, a flume of 20 m by 3 m and 0.4 m deep that tilts periodically to drive reversing tidal currents (Kleinhans et al., 2017). The tilting causes similar sediment mobility of coarse sand to nature

(Kleinhans et al., 2017). The tidal forcing was kept simple, with one primary tidal component only, and a small river discharge. In all experiments we supplied a minor river discharge and applied monochromatic waves by a paddle in front of the ebb delta to diffuse coastal transport and form beach ridges (as in Leuven et al., 2018c). The first experiment had only sand, while the second and third had mud or live vegetation to compare the two styles of floodplain formation against a control without floodplain. Low-density sediment was supplied as a mud simulant; this experiment is also reported in Braat et al. (2019). The

experiment with live vegetation reported here is novel and is based on vegetation experiments in van Dijk et al. (2013) and Lokhorst et al. (2019). Furthermore, to avoid practical difficulties with flow measurements, we apply a numerical flow model to all experiments (Weisscher et al., 2020), modifying the roughness where vegetation is present. In this section, the experimental methods, vegetation treatment and numerical modelling methods are described.

### 2.1 Experimental methods

Experiments were conducted in the Metronome, which tilts with a sinusoidal motion over the short middle axis with a slope amplitude of 0.0068 m/m at a period of 40 seconds. This drives tidal currents with a depth-averaged velocity amplitude in the channels of about 0.3 m/s.

The initial bed was screed flat at millimetre accuracy. The initial convergent channel of 0.2-1.0 m wide and 0.03 m deep was carved into a sand bed of 0.07 m thick, while the sea was kept free of sand between 17.8–20 m. The initial water depth was

0.025 m. We used poorly sorted sand with a $D_{10}$ of 0.33 mm, a $D_{50}$ of 0.57 mm and a $D_{90}$ of 1.2 mm. Flume settings were chosen such that sediment mobility, expressed as the Shields number, was about 0.5 (Kleinhans et al., 2014, 2017; Braat et al., 2019).

The landward boundary condition is a river discharge of 0.1 L/s supplied for half the tidal cycle when the flume was tilted seaward. The river discharge in isolation did not mobilise the sediment but was essential to maintain a long estuary (see online





supplement in Braat et al., 2019). The seaward boundary condition is a broad-crested weir with constant water supplied from a basin to have a constant head condition. The weir moves up and down in the opposite phase of the tilting motion such that the sea remains approximately horizontal. This prevents surges into the estuary and draw-down that would cause incision in the ebb delta and ensures that the tidal flow is entirely forced by the periodically varying gradient. The weir amplitude was reduced linearly with the progradation of the delta towards the boundary. Furthermore, monochromatic waves of 1 s

period and 0.006 m height were generated by a horizontal paddle during the landward tilting half of the tidal cycle. Waves in isolation hardly mobilised the sediment but in combination with tidal currents caused coastal diffusion to round the delta during progradation. Testing and analyses of scaling of waves is described in the supplement to Leuven et al. (2018c).

    Bathymetry was measured every 1,000 cycles (see Braat et al., 2019, for slight deviations in timing) on the dry bed by stereophotography and structure from motion by Agisoft PhotoScan, constrained by control points at millimetre accuracy (also

see Leuven et al., 2018c; Braat et al., 2019). Vegetation and mud were recognised in the overhead imagery through color (also see van Dijk et al., 2013; Braat et al., 2019).

## 2.2    Floodplain formation

One experiment had a supply of suspended sediment sufficient to simulate the formation of mudflats. Mud was simulated by a supply of crushed nutshell at the river boundary. This experiment has already been published earlier (Braat et al., 2019) and is

reanalysed here. The nutshell had a diameter of 0.2 mm and a dry density of 1350 kg/m$^3$. The nutshell was kept in suspension in a mixing tank and was supplied at a dry volume of 0.001 L per tidal cycle, so that a total volume of 0.013 m$^3$ was added to the experiment. Given a 20 s river discharge duration per tidal cycle, the mud concentration in the river influx was about 810 mg/L. The velocity required to suspend the mud is much lower than that to suspend the sand, so that the mud can deposit in much shallower flow (Braat et al., 2019).

Another experiment had regular vegetation recruitment events through periodic supply of vegetation seeds of two species followed by four days of rest to allow sprouting above the still water surface. This was started after 4,500 cycles and done every 2,000 cycles. Vegetation was added by seeding at the river boundary, similarly to the hydrochorous seed distribution in the river experiment of van Dijk et al. (2013) (also see for scaling of vegetation Kleinhans et al., 2015a). After growth tests, we selected two species that grow above the water surface to simulate effects of salt marsh species by enhanced flow resistance

and capturing of suspended sediment Lokhorst et al. (2019). *Veronica beccabunga* forms 10 mm tall plants that grow at the waterline in dense elongated patches and above the waterline in sparse cover, and *Lotus pedunculatus* grows up to 20 mm tall at somewhat higher elevations above still water, usually with sparse cover and sometimes in tussocks. Both species stopped growing after 5-7 days as the sand and tap water were free of nutrients. Chlorine was added to the water to prevent algae and bacterial growth, and the bed was treated with anti-algae spray at 8,000, 10,500 and 12,500 tidal cycles.

All seeds were soaked for 24 hours to prevent floating and speed up germination. We released batches of seeds at the upstream boundary after every dry bed photograph. After 10 tidal cycles for initial wetting, 12.5 g or about 10,000 seeds of *Lotus pedunculatus* were supplied. Another 25 cycles later, 3.75 g or 15,000 seeds of *Veronica beccabunga* were allowed to disperse for 35 more tidal cycles and then left to germinate for four days without tilting but with river discharge. These 70 tidal





cycles were subtracted from the 1,000 cycle run before the next dry bed photography session so that spacing between the DEMs
was exactly 1,000 cycles. Laboratory conditions were 300 lux light intensity at all times, water temperature of 20° C and a
room temperature of about 17–20° C. The duration of the vegetated experiment was about two months.

## 2.3  Numerical flow model

The numerical flow model Nays2D solves the shallow water equations to predict two-dimensional, depth-averaged flow ve-
locity, water surface elevation and water depth in small-scale systems such as flumes, and was adapted to drive the flow by
tilting as in the Metronome (Weisscher et al., 2020). Previous results showed a good similarity between modelled flow and
flow velocity and depth measured by surface particle imaging velocimetry and water colour-derived depth (Weisscher et al.,
2020). This code was applied to the measured bathymetries and applied boundary conditions of the three experiments. The
output of water depth and flow velocity was calculated at a 2.5 by 2.5 cm resolution. For the sand and the mud experiments,
the measured bed surface elevation is taken with a spatially constant surface Manning roughness of 0.02 s m$^{1/6}$.

For the vegetated experiment, spatially variable roughness is applied depending on the estimated vegetation density. Vegeta-
tion was filtered off the bed surface elevation map and assigned the same roughness as the other two experiments in unvegetated
areas. Vegetation roughness was assumed to be related to plant stem density and diameter. Earlier experiments demonstrated
that, under similar conditions, the flow velocity halved and the roughness doubled for a vegetation density of 2 plants per
cm$^2$ (Lokhorst et al., 2019). Here, the roughness is calculated from estimated stem density and known drag coefficient and
stem thickness using the Baptist et al. (2006) relation for emergent vegetation, which was in good agreement with data for the
species and conditions of the experiments (Lokhorst et al., 2019). A Manning coefficient of 0.1675 s m$^{1/6}$ was calculated for
the densest vegetation with 20 stems/cm$^2$ and a stem diameter of 0.5 mm. Based on this, a roughness map was created for each
timestep based on observed vegetation density (example in Fig. 7).

## 3  Results

### 3.1  Morphodynamics pattern and development

All three experiments developed a convergent, multi-channel estuary with mid-channel bars and a prograding ebb delta (Fig. 2,
movie in Supplementary Online Materials). All three estuaries widened by bank erosion from the initial, monotonously con-
verging estuary to form channels and bars. As the original high sand bed flanking the estuaries was never flooded, mud or
vegetation was confined in the reworked area. The vegetation and the mud first settled on the most upstream mid-channel
bars and the shore-connected bars (Fig. 2b,c). Mud and seeds were also observed in suspension in the channels and deposited
seaward of the ebb delta (without sprouting).

The distribution of bed elevations generally broadened in the experiments (Fig. 3). Initially, the carved channel rapidly
shallowed by deposition of sediment eroded from the banks. After about 1,000 cycles, the channel-bar morphology had formed
and the slower process of estuary widening dominated the trend in bed elevations. Especially in the middle reach (Fig. 3b),





the muddy shoals increased several millimetres in elevation, while the channels deepened. All three estuaries gained a broader bed elevation distribution with deeper channels and higher shoals in the middle reach, as well as in the upstream reach for the vegetated estuary. In the other reaches, the bed elevation distributions did not broaden much. In the upstream reach (Fig. 3a), the channels became shallower for the sandy estuary and stayed constant for the muddy estuary, while the vegetated estuary developed slightly deeper channels. In the downstream reach (Fig. 3c), bed elevations increased particularly in the vegetated

estuary, while the sandy estuary remained the deepest.

The channel-bar patterns and bed elevation distributions were caused by the morphodynamics of channel erosion and migration, estuarine bank erosion, and bar formation and accretion. In the upstream and middle reaches of the muddy and vegetated estuaries, these processes reduced considerably compared to the control experiment (Fig. 4). As a result, the upstream 7 m of the estuaries with floodplain remained narrower, especially of the muddy estuary. The downstream half of the estuaries

(10–16 m) widened similarly but upstream the muddy estuary showed fewer channels (Fig. 4) and less lateral channel migration (Fig. 5) than the vegetated estuary and the control. The vegetated estuary was intermediate both in channel carving and migration (Figs 4,5).

The differences in dynamics are also visible in the virtual stratigraphy of the cross-sections (Fig. 6). The cross-sections of the sandy control show rapidly varying ages of deposits, indicating perpetual avulsion of channels. The vegetated estuary tends

to channel migration while the muddy estuary had predominantly vertically accreting bars, showing that the mud was more effective at reducing bank erosion and lateral channel migration than the vegetation. This is surprising as the rooting depth of the vegetation is approximately the same as the vegetation height and the typical channel depth (Lokhorst et al., 2019), while the mud layer thickness is an order of magnitude smaller (Braat et al., 2019).

### 3.2  Hydrodynamics: flow and tidal prism

The hydrodynamics were characterised by numerical flow modelling for the entire tidal cycle on all measured bathymetries. At the end of the experiments, the path of maximum ebb velocity during the tidal cycle mainly follows a single channel in all three experiments, while the maximum flood velocity is more distributed over the width in the downstream half of the estuaries (Fig. 8). The velocities in the upper reach are generally lower in the muddy estuary than in the sandy estuary, which is consistent with the lower width and the raised bars due to mud deposition. The vegetated estuary shows a striking difference

with the other two: the higher friction of vegetated bars (compare Fig. 7 and Fig. 8) not only strongly reduces flow velocity over the bars, but also focuses the flow into the channels, even during the flood phase when the flow enters the system through the relatively wide mouth unconfined by vegetation.

To quantify the velocity patterns in the different experiments, the probability distributions of the flow velocities were calculated for the shallowest and the deepest parts in middle section, halfway and at the end of the experiments (Fig. 9, consistent

with the bed elevation percentiles shown in Fig. 3b). Comparison between 6,000 cycles and 13,000 cycles shows a reduction of flow velocity over the shoals (peaks of dashed lines shift to the left from Fig. 9a to c and b to d), especially for the vegetated experiment. Comparison between the experiments at 6,000 cycles shows that the ebb and flood flow in the channels is lower and broader distributed in the muddy estuary (Fig. 9a,b), although the differences disappear towards the end of the experiments.





The vegetated experiment, on the other hand, has a narrower distribution of high channel velocities around 6,000 cycles, and

lower velocities over the shoals around 13,000 cycles when vegetation has settled more generally. Clearly, the vegetation baffles the flow by resistance on the shoals, whereas the mud reduces the flow in the channels.

The tidal prism is here not only calculated at the mouth but at every cross-section along the estuary to resolve more local effects of vegetation and mud (as in Braat et al., 2019). The tidal prism increased in the seaward direction and is the smallest for the experiments with mud (Fig. 10). After 13,000 tidal cycles the tidal prism in the mouth area had reduced most for

the muddy experiment compared to 6,000 cycles. In the upstream half, the change was small though the tidal prism of the sandy experiment continued to increase. This is consistent with the rapid initial channel shallowing of the upstream reach (Fig. 3) and concurrent infilling (Fig. 6). The spatial variation in tidal prism shows the expected seaward increase, but has a superimposed secondary pattern of around 6 m and 14 m. A similar pattern in the tidal prism for the initial bathymetry (dashed line in Fig. 10) shows that this is likely due to a deviation of tidal flow generation by tilting from what is generally observed in

natural estuaries. Specifically, in somewhat more open and less filled reaches of the estuaries, the tilting was visually observed to drive periodically reversing flow regardless of the connection with the mouth. Nevertheless, the landward penetration of tides reduced over time. The most upstream flood flow velocity was slightly increased in the muddy estuary as the flow was concentrated in a single channel and flanked by high mudflats, but the tidal prism at that point was already smaller than in the other estuaries.

The tidal prism changed through time in different ways along the estuaries (Fig. 11). In the upstream reach, the tidal prism of the muddy estuary decreased rapidly in the beginning and then remained about constant, while that of the sandy and vegetated estuaries decreased almost linearly over time. In the middle reach, the tidal prism in the muddy and vegetated estuaries stay about constant after initially increasing, while that of the sandy estuary continues to increase nearly until the end. Close to the mouth, the tidal prism is the collective result of tides along the entire estuary (excluding a secondary effect of locally generated

tides due to tilting) and the differences in magnitude are largest. Here, the sandy estuary has the largest tidal prism and the muddy estuary the smallest by about two-thirds, and the vegetated estuary takes an intermediate position. The muddy and sandy estuaries initially show a rapid increase of tidal prism but then a gradual, though limited, decline. This is consistent with the declining depth in the downstream reach (Fig. 3c).

The reduced tidal prism in the muddy estuary is not merely lower than in the other two experiments because a certain volume

of mud was supplied that filled space: the total added volume was 0.013 $m^3$, which is only a small fraction of the difference in tidal prism between the sandy and the muddy estuary (about 0.5 $m^3$ during nearly the entire experiment; see Fig. 11). As also shown in Figs 4 and 5, mud had a system-wide morphological effect on the estuary development. Regardless of the limited filling, the mud reduced the over

The vegetated estuary shows a pattern of gradual increase of tidal prism until about 8,000 cycles followed by a decrease,

unlike the muddy estuary. This can also be seen in the higher channel velocities in the vegetated experiment (Fig 9a,b). The decrease is due to the gradual increase in vegetation cover as the recruitment events added up, and the particularly rapid expansion of vegetation after about 8,000 cycles (Fig. 12). As a result, the tidal prism reduction in the vegetated estuary is approaching that of the muddy estuary after 13,000 cycles (Fig. 10). It is unclear whether the muddy estuary would have filled



further, given the mud supply, or whether the vegetated estuary would have been covered more extensively by vegetation with
25  the ongoing seed distribution events. While the tidal prism at the end of the experiments was nearly constant, the erosion of the
outer estuary banks, composed of pure sand, and the expansion of the ebb delta could slowly continue.

## 4  Discussion

The experimental results suggest that vegetation and mud have similar, but not the same, effects on the landward reduction
of tidal energy. While mud fills the accommodation space to reduce lateral channel mobility and overall depth, the vegetation
reduces the tidal flow through vegetation-enhanced hydraulic resistance on the bars. Surprisingly, the enhanced bar accretion
in the muddy estuary reduced the flow over the bars less than the vegetation despite the blockage of flow by mudflats growing
up to the water surface. This is all the more surprising as the shallow experimental flows were probably laminar for a greater
part of the tidal cycle (Kleinhans et al., 2014, 2015a, 2017) which would increase friction more.

Equally surprising is the much lower lateral channel mobility in the muddy estuary than in the vegetated estuary, despite the
fact that the mud deposits were much thinner than the channels (Braat et al., 2019), which allows for unhindered undercutting
into the noncohesive sand underlying the mud. Moreover, the mud simulant, nutshell, was not found to be so cohesive as to
reduce bank erosion, even after tens of days (Fig. 13 in Braat et al., 2019). In contrast, the vegetation rooting could be as deep
as the channels (Lokhorst et al., 2019) but this did not reduce the mobility as much as the mud. The growth of vegetation in
nature, even at supratidal level, causes high hydraulic resistance during high tides. This leads to reduced and often negligible
flow on the bars and strong focusing of the flow in the channels as also found in experiments and models of river systems (Tal
and Paola, 2009; Braudrick et al., 2009; van Dijk et al., 2013; van Oorschot et al., 2016; Kleinhans et al., 2018). This happens
particularly in upstream reaches of the estuaries where the tidal dynamics are reduced sufficiently for vegetation to settle and
where estuarine bars and mudflats can accrete to the high-intertidal and supratidal levels required for vegetation to settle (Vos
and van Kesteren, 2000; Woodroffe et al., 2016; Lokhorst et al., 2018). Possibly, the difference between the vegetated and
muddy estuary occurred because the cover of vegetation is smaller than that of the mud, and given more time for recruitment,
or more inundation-resistant species, the vegetated estuary might have developed more similar to the muddy estuary.

A morphological effect of filling of intertidal space by vegetation and by mud is the reduction of flow shear stress over tidal
bars. This, combined with the increased resistance against erosion, diminishes the likelihood of cross-cutting bars by channels
that would otherwise lead to channel braiding as in rivers (Ashmore, 1991). The data suggest a transition from rapid channel
avulsion in the control experiment to more gradual migration in the vegetated estuary and even reduced migration in the muddy
estuary. This filling effect is paralleled in river morphodynamics, where initiation of braiding through chute cut-offs is inhibited
by the vegetation and the suspended sediment on the floodplain. Both vegetation and suspended sediment reduce excess shear
stress, regardless of whether the sediment is cohesive (van Dijk et al., 2013) or non-cohesive (Braudrick et al., 2009; van Dijk
et al., 2013). This reduced channel cutting tendency is important for overall floodplain formation because channel migration
is far more effective in eroding bars and removing floodplain by undercutting than direct overflow during flooding. This was
also evident from the comparison between the vegetated and muddy estuaries with the control. In field data, the tendency to





stabilise channels and reduce bar cross-cutting was also observed (Swinkels et al., 2009; Wang et al., 2015; van Dijk et al., 2021) but the effects of increasing sand bar height, mud sedimentation and salt marsh expansion occurred simultaneously and could not be separated, unlike in the experiments.

Moreover, channels in this kind of estuary are rather fixated in position by topographic forcing around the mid-channel bars (Leuven et al., 2018a), by embankments and by dredging (van Dijk et al., 2021), which nearly completely removes the tendency of floodplain destruction by lateral bank undercutting. This was also observed in earlier experiments (Leuven et al., 2018c; van Dijk et al., 2021). As floodplain accretion continues unhindered, cohesive sediment and plants also reduce potential erosion by overflow (Brückner et al., 2020). Where the more protected flanks of the estuary fill up with fine sediment and vegetation,

the tidal system is effectively constrained and reducing in dimensions as observed in reconstructions (Vos and van Kesteren, 2000; Woodroffe et al., 2016; de Haas et al., 2018) and models (Lokhorst et al., 2018; Braat et al., 2017). This contrasts with idealised modelling and with experimental conditions with cohesionless sediment only so that tidal systems continue to expand (Zhou et al., 2014; Kleinhans et al., 2015b). The only factor reducing the tidal penetration in such estuaries is tidal damping by the bottom friction, which increases with reducing depth and shallow bar area, and tidal damping by large flood storage

areas, which were absent in the experiments. Friction increased more rapidly in the muddy estuary due to the upstream depth reduction and in the vegetated estuary due to the large relative surface area with vegetation (Fig. 12).

    The two local effects outlined above, namely space-filling and flow resistance, could both lead to lateral constraints and tidal channel deepening, which would, according to the first hypothesis, enhance tidal energy in the upstream estuary and cause further landward tidal penetration. However, the long-term development of tidal prism in the experiments demonstrate a

positive feedback between floodplain formation, channel morphodynamics and planform dimensions. The filling of intertidal space, on the fluvial-tidal transition and further seaward, causes a reduction in the tidal prism at the mouth as hypothesised in de Haas et al. (2018). This is especially the case for the muddy estuary where accommodation space is filled with mud, and less so for the vegetated area where flow over vegetated bars is reduced. Regardless of the widening by cohesionless bank erosion, the tidal prism at the mouth reduced due to vegetation and mud and, in the sandy estuary, also due to shallowing. In this sense,

the effect of floodplain formation on an estuary is different from that on a river, where the flow discharge is locally unchanged, as it is externally imposed. As a result, narrowing of the channel in rivers leads to deepening at the same time, which is not the case in estuaries where the tidal discharge reduces.

    The long-term effects of a reducing tidal prism on the system are considerable. Where a large intertidal area with delayed outflow would have enhanced ebb-dominance and sediment export, its reduction due to floodplain formation may invert this

tendency and cause import (Friedrichs and Aubrey, 1988; de Haas et al., 2018). In other words, mud deposition and vegetation development could cause large-scale filling of estuaries. This direct link between sedimentation and vegetation growth and estuary dimensions dominates over the effect of floodplain observed in rivers because the total flow discharge is modified in estuaries as opposed to rivers. As filling progresses, this zone is expected to shift seawards as well, but testing this requires experiments with initially unfilled estuaries and control on the landward sediment transport from the coastal zone. These

findings explain how initial estuary widening, following catastrophic ingression (de Haas et al., 2018), can be followed by constraints and even filling.

In nature, the calm hydrodynamic conditions could be conducive to organic matter production and storage leading to organogenic rise of the bed surface in salt marsh (Kirwan et al., 2016) and mangrove (Woodroffe et al., 2016), especially once removed further from the active estuary where precipitation leads to freshwater conditions. The latter is more likely to occur upstream in the estuary, which is consistent with the observation that Holocene tidal systems filled in the seaward direction (Vos and van Kesteren, 2000; Woodroffe et al., 2016; de Haas et al., 2018). On the other hand, past numerical modelling showed that the reduced flow through the canopy also reduces the sedimentation within the vegetation Brückner et al. (2020). This could not be tested well in the experiments, because the flow through the experimental vegetation becomes laminar and would not suspend sediment Lokhorst et al. (2019). Regardless, the collective effect of the different mechanisms of mudflat and salt marsh formation is to constrain estuary dimensions and, given sediment input, the forming of land.

## 5  Conclusions

Mudflats and salt marsh development in natural, multi-channel estuaries reduce system dimensions and dynamics similarly to floodplains in rivers, but mud and vegetation have subtly different effects. While both mud and vegetation reduce the overall tidal prism along the estuary and the landward penetration of tides, vegetation effectively concentrates tidal flow in channels while mud effectively reduces lateral channel migration. Both mud and vegetation likely contributed to the lateral constraining of estuaries initially formed by catastrophic ingression. mudflat sedimentation and salt marsh expansion also have indirect effects on the tidal dynamics, which lead to a positive feedback of enhanced filling through the reduction in tidal prism; an effect absent in rivers where flow discharge is mostly independent of the upstream morphology.

*Data availability.* Bathymetries (in Matlab and in NetCDF4 format and numerical flow model input and output data are made available at Yoda https://doi.org/10.24416/UU01-R3ZUC9

*Video supplement.* video of overhead imagery for all three experiments

*Author contributions.* M.G.K. conceived the study, built the laboratory facility and wrote the manuscript. L.R. made most of the figures. L.R., I.R.L. and L.B. conducted the experiments supervised by M.G.K. S.A.H.W. and L.R. set up and conducted the numerical models with help of S.A.H.W., and all authors contributed to idea development and manuscript preparation.

*Competing interests.* The authors declare no competing interests.



*Acknowledgements.* Reviewers will be acknowledged. Funded by the Netherlands Science Foundation NWO-TTW Vici grant 016.140.316/13710 to M.G.K. and the ERC Consolidator grant 647570 to M.G.K. Technical support of the experiments by the lab technicians and help during experiments by Jasper Leuven are gratefully acknowledged. Discussion with Harm Jan Pierik and Tjalling de Haas helped to improve the paper.



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

Earth **Surface** Dynamics
Discussions

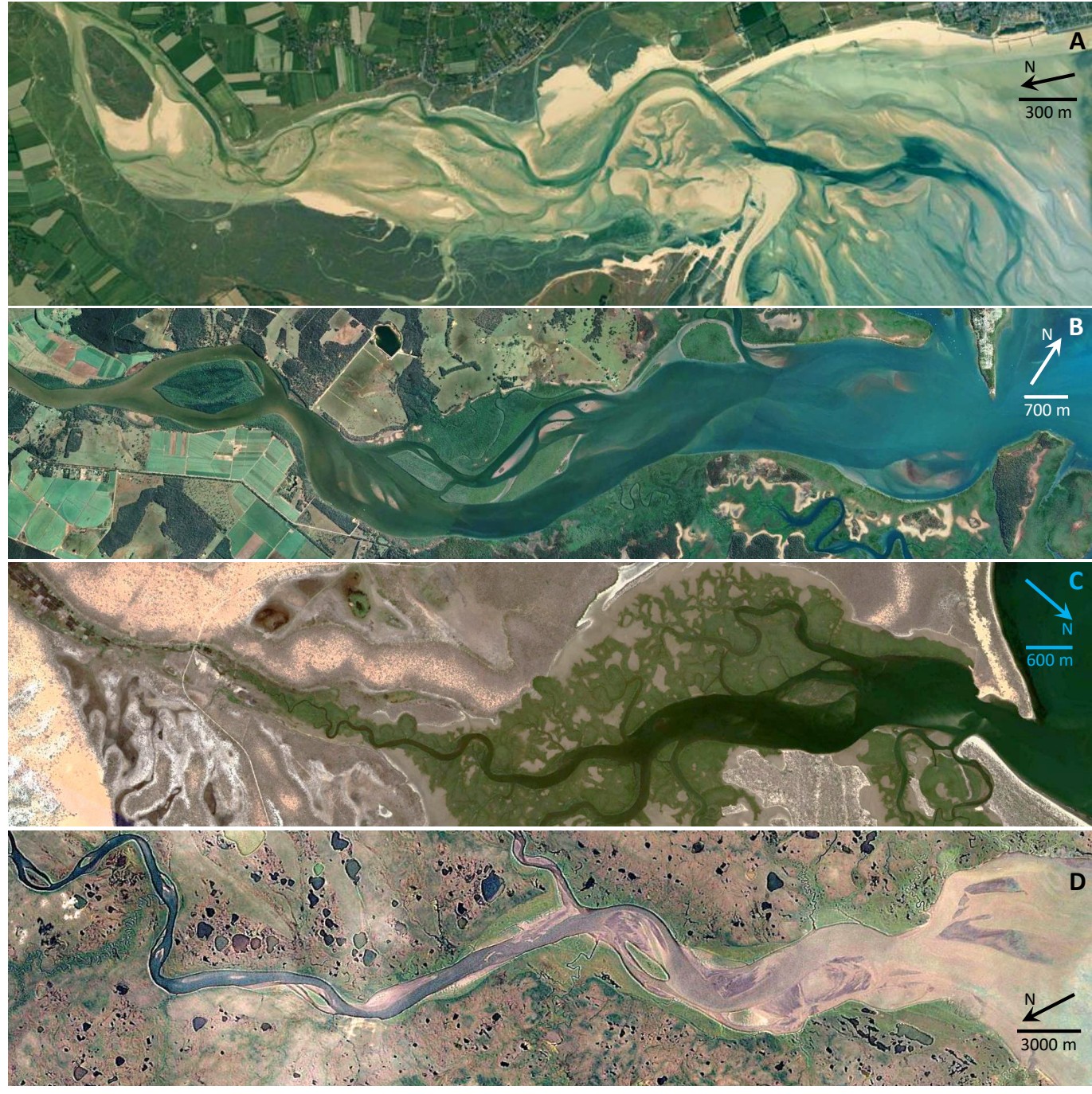

**Figure 1.** Example estuaries with mid-channel bars that have vegetated flanks and bars. (a) La Sienne estuary, W Normandy, France. May 2010. (b) Mary river, SE Queensland, Australia. Note upstream increasing sediment concentration and mudflats near the mouth. Image of 6 July 2021. (c) Mangolovolo estuary, SW Madagaskar. Image of September 2016. (d) Levelock creek, Alaska. Note extensive muddy bars. Image of December 1999. All images from ⓒ Google Earth accessed 5 August 2021 and contrast-stretched as a whole.

Earth **Surface**
Dynamics
Discussions
EGU



**Figure 2.** Bathymetry after 13,000 tidal cycles (see movie in Supplementary Online Materials). (a) Control experiment with only sand. (b) Estuary with two vegetation species, cover indicated in green. For visualisation, the binary map of vegetation was dilated by a kernel of 4 mm in Matlab. (c) Estuary with mud simulant, cover indicated in black (low mud experiment from Braat et al., 2019).



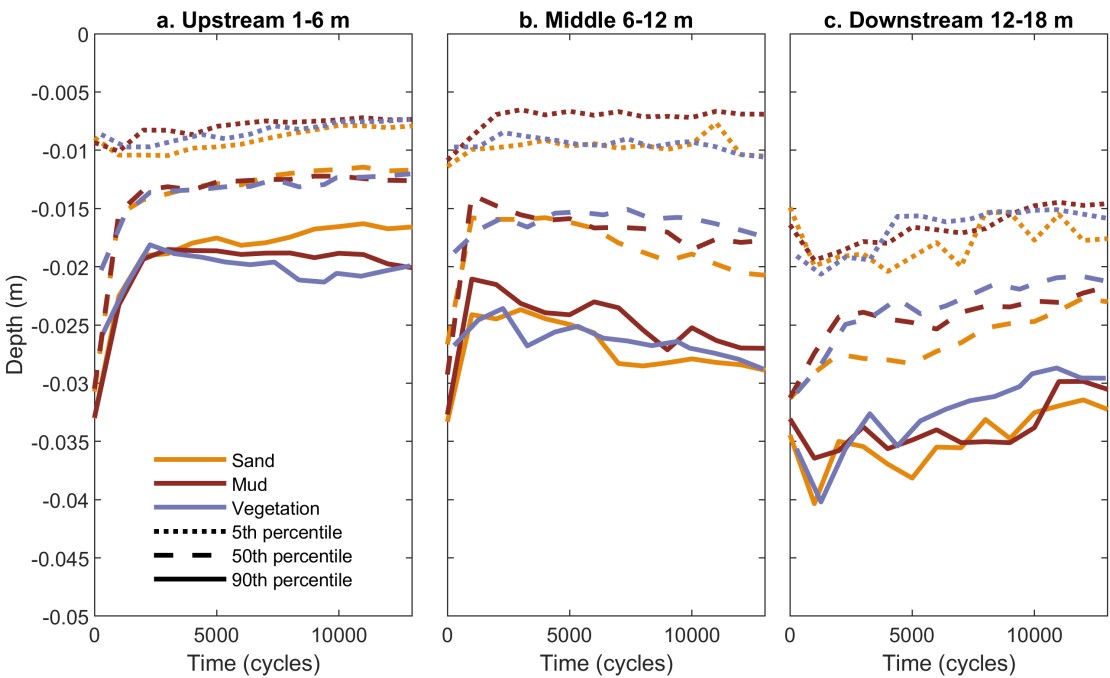

**Figure 3.** Time series of estuary depth below initial bed level, subdivided into an (a) upstream part (1-6 m), a (b) middle part (6-12 m), and a (c) downstream part (12-18 m). The 5, 50 and 90 percentiles of depth represent the shoal depth, the median depth and the main channel depth.

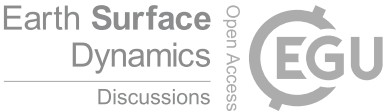

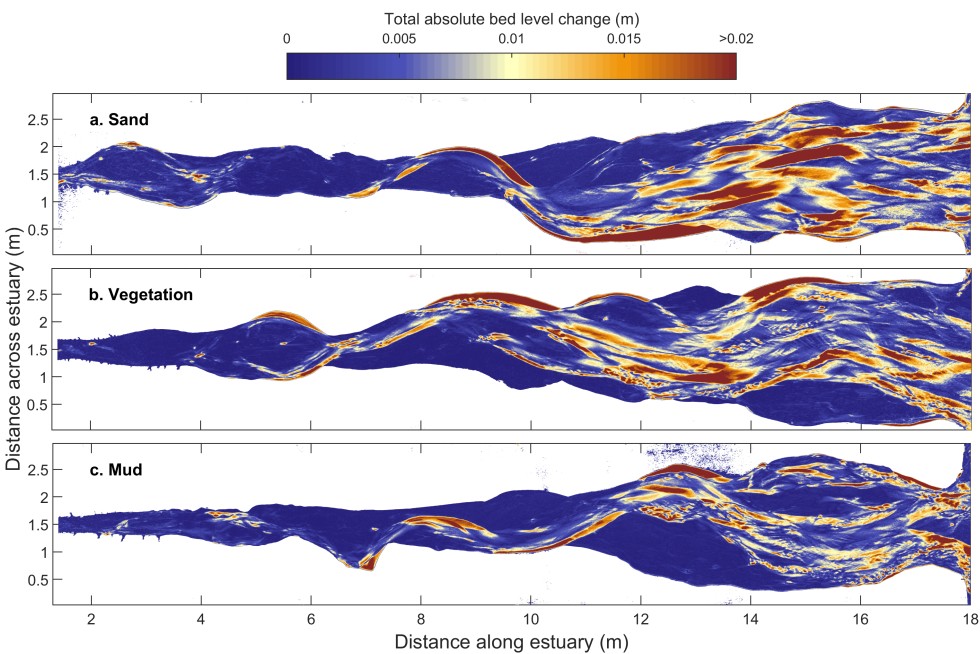

**Figure 4.** Morphodynamics characterised by cumulative absolute bed level change over the duration of the experiments with (a) sand only, (b) vegetation and (c) mud. To avoid overrepresentation by the early stages, only data of every 1,000 cycles was used. The absolute bed level change is calculated from the moment that an area became part of the estuary for the first time.



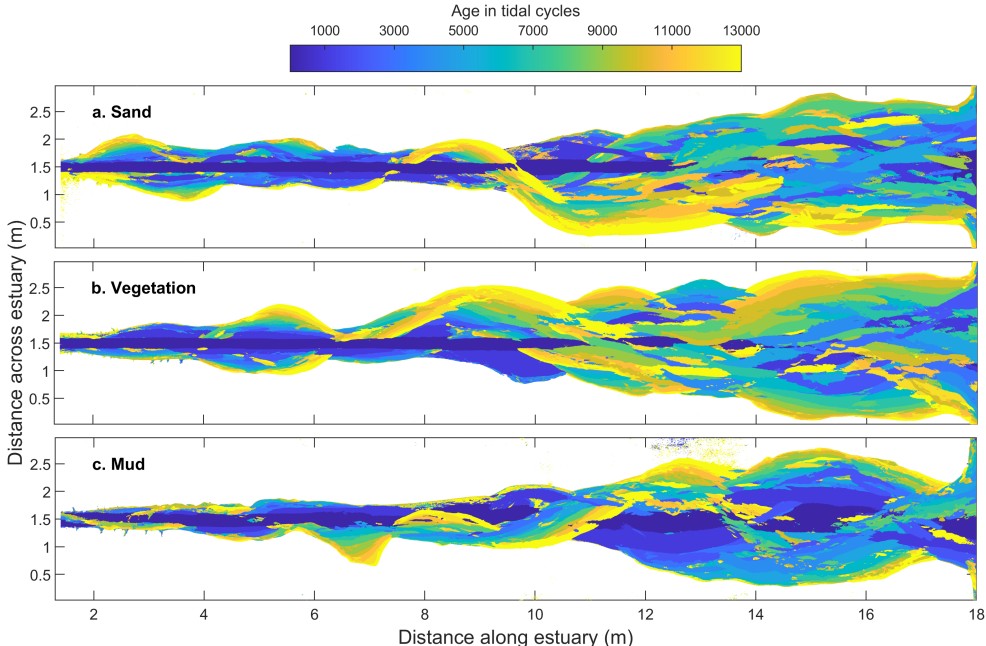

**Figure 5.** Morphodynamics characterised by the age of the largest depth reached over the duration of the experiments with (a) sand only, (b) vegetation and (c) mud. To avoid overrepresentation by the early stages, only data of every 1,000 cycles was used.





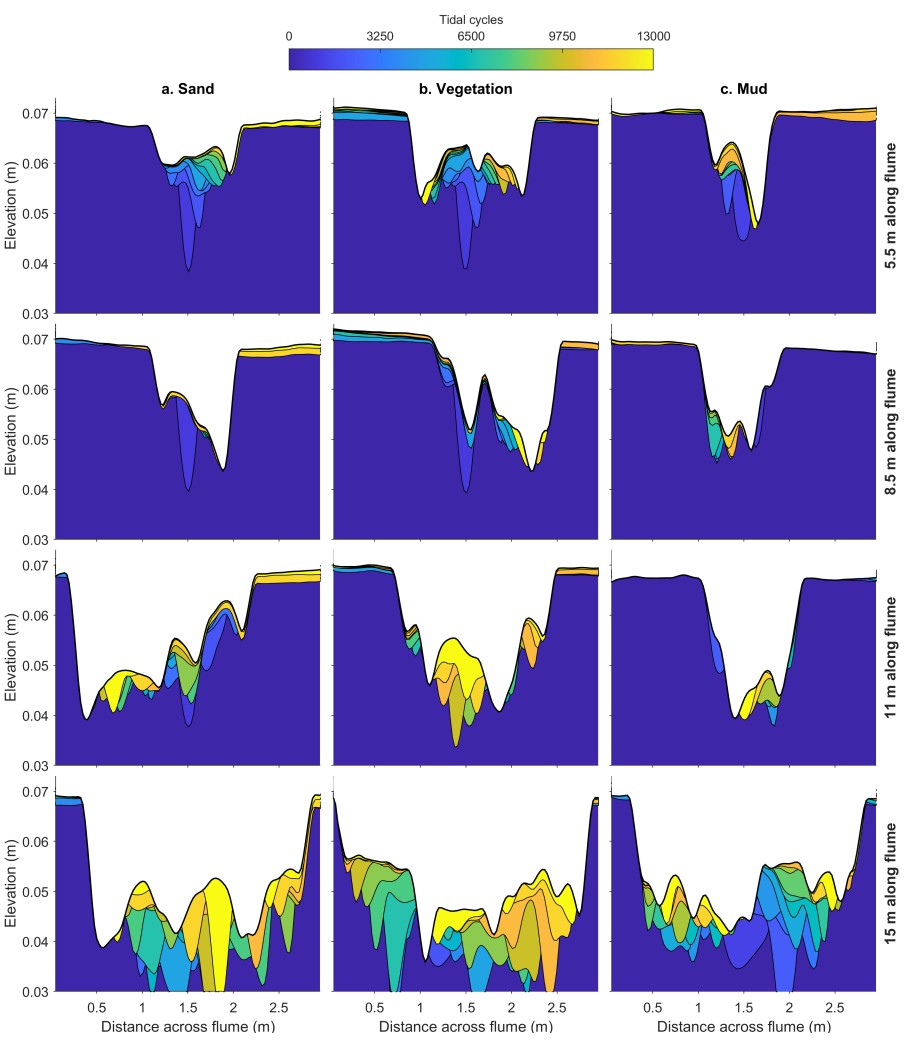

**Figure 6.** Cross-sections with age of deposit, showing the preserved part of the morphodynamics in the experiments with (a) sand only, (b) vegetation and (c) mud for four positions along the flume. Bed level data of every 1000 cycles were used. The vertical exaggeration is a factor of 50.



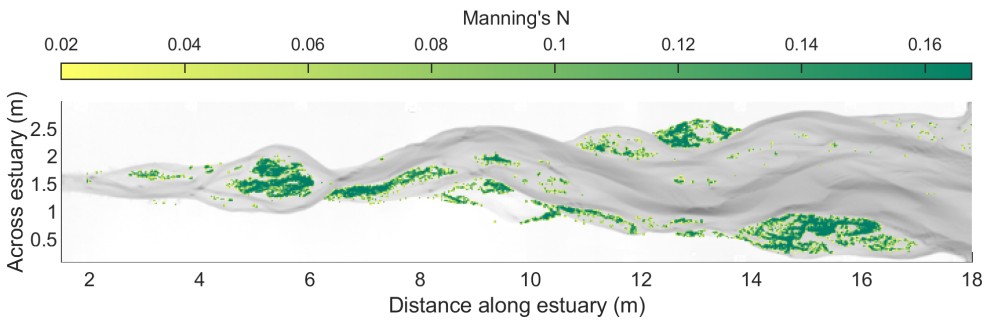

**Figure 7.** Example of Manning roughness values for the vegetated estuary specified as initial condition in the Nays2D flow model at 13000 tidal cycles. The Manning roughness of the unvegetated bed is set to 0.02 in the model.

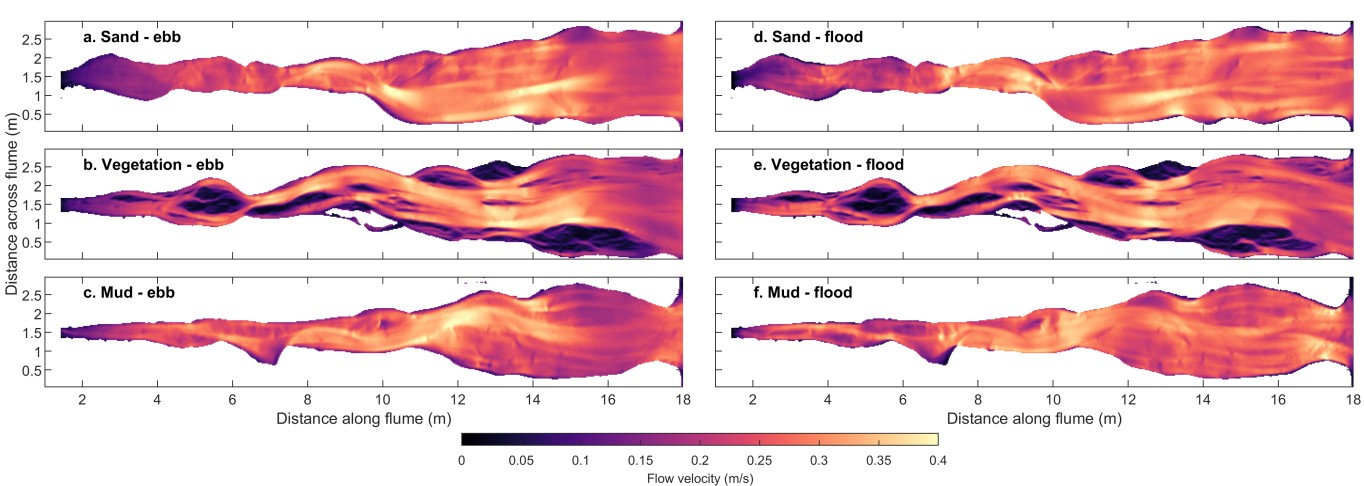

**Figure 8.** Maximum flow velocity calculated with the flow model for the end of the experiment during ebb (left) and during flood (right) in the experiments with (a) sand only, (b) vegetation and (c) mud.

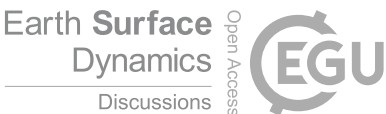

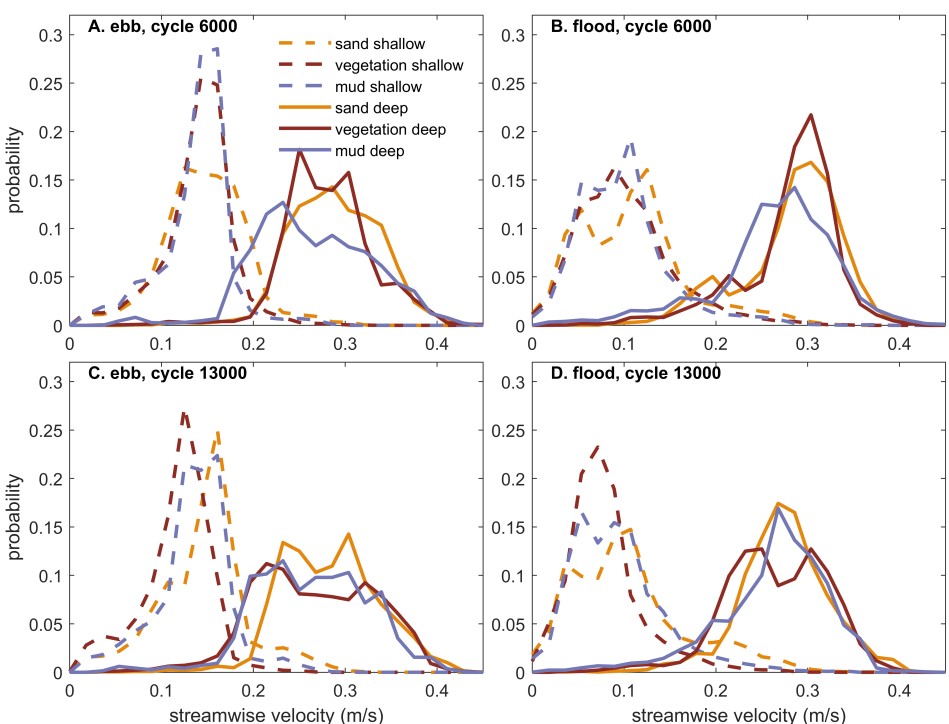

**Figure 9.** Distributions of maximum flow velocity calculated with the flow model for the middle (top; a,b) and end (bottom; c,d) of the experiment during flood (left; a,c) and during ebb (right; b,d). The selected depths ranges are shoal areas (0>depth>-0.015 m) and channels (-0.025>depth>-0.050 m) for consistency with Fig. 3b.

Earth **Surface**
**Dynamics**
Discussions

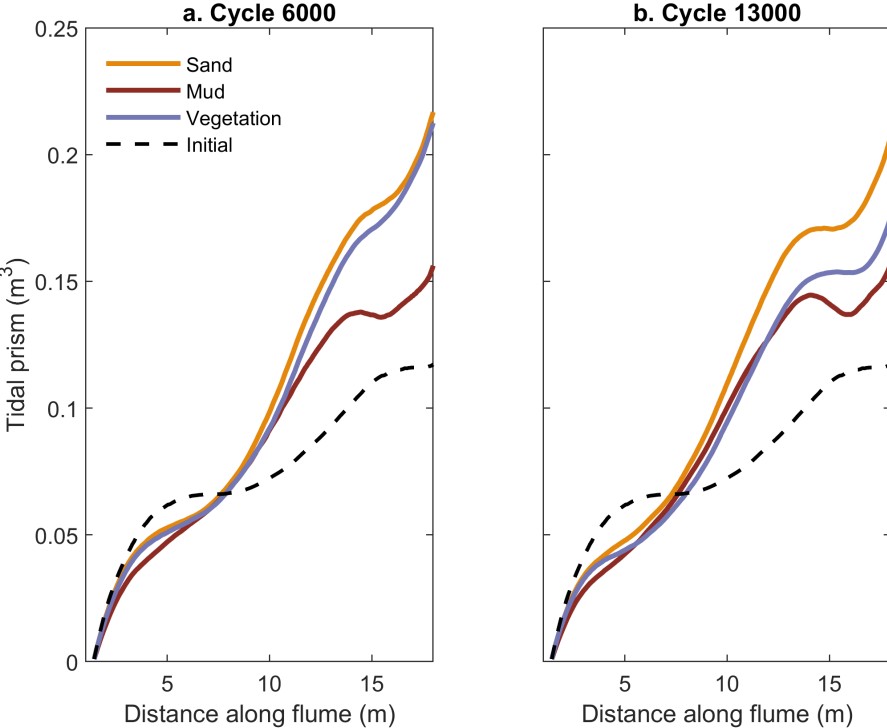

**Figure 10.** Tidal prism along the experimental estuaries calculated with the flow model after (a) 6,000 and (b) 13,000 tidal cycles. The estuary mouth is located at 17.8 m.



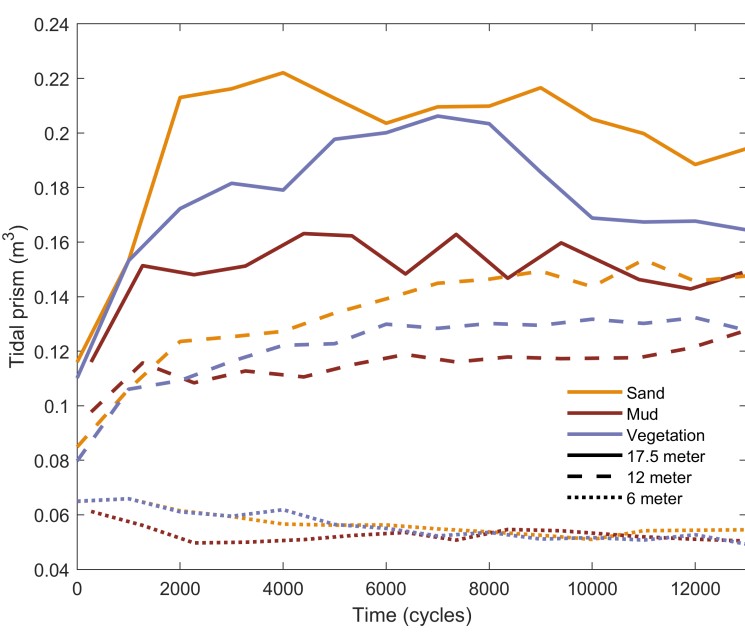

**Figure 11.** Timeseries of tidal prism calculated along the estuaries at distances of (a) 6 m, (b) 12 m and close to the mouth at (c) 17.5 m.

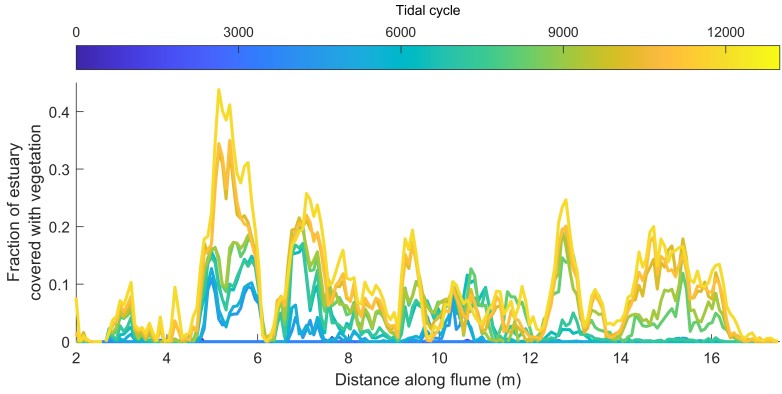

**Figure 12.** Development of vegetation cover as a fraction of estuary width along the estuary.