# Peer review of "Estuarine morphodynamics and development modified by floodplain formation"

_Earth Surface Dynamics, 2021_

## Author Response (AR1)

We thank the reviewers for their time and constructive comments. Below we
respond point by point and indicate the changes made in the manuscript.
Maarten Kleinhans, on behalf of all authors

RC1: William Kearney, https://doi.org/10.5194/esurf-2021-75-RC1
General comments

The authors present the results of laboratory experiments that reproduce the
development of floodplains within estuaries. They compare a control case with
only sand to one with mud and one with live vegetation, which enables them to
show subtle differences in the way mud and vegetation affect floodplain
formation. Their conclusion that mud affects tidal propagation in the estuary
by filling accomodation space while vegetation imposes additional hydraulic
resistance certainly makes sense and is well supported by their data. The
vegetated estuary starts off more like the sandy estuary and becomes more like
the muddy estuary as vegetation establishes. The main limitation of their
experiments seems to be that the experiments were not run long enough to
determine if the vegetated and muddy estuaries will completely converge. These
experiments provide an important complement to field and numerical studies of
estuarine floodplain formation, and this manuscript is an excellent
description of the experiments.

REPLY Indeed the experiments did not run long enough for complete convergence,
but as with any process that has an exponentially decaying rate of change this
would have taken a prohibitive time period as well as have induced growth of
pests in parts of the sediment without significant flow (especially in buried
nutshell or decaying vegetation, both of which were very limited now), despite
pest control.
In Results Section 3.2 on the hydrodynamics we now explain that convergence is
not complete. In the discussion we already stated that "given more time for
recruitment, or more inundation-resistant species, the vegetated estuary might
have developed more similar to the muddy estuary".
The reviewer agrees with us that the experiments are nevertheless interesting
complements to modelling and field studies, and that the numerical flow
modelling used here is useful and not critically limited, while the comparison
between the experiments is the most informative aspect.

Specific comments

I was somewhat skeptical of using a numerical model to replace measurements of
hydrodynamics, especially when it comes to the effects of vegetation on the
flow, but the citations to Weisscher et al. (2020) and Lokhorst et al. (2019)
suggest that this is not unreasonable. In any case the authors'
interpretations do not rely too heavily on the exact flow velocities achieved
in the different experiments, and the tidal prism estimates are probably more
robust to model inaccuracies.

REPLY We now state more clearly in the introduction to the model in the
Methods section that this model has been applied successfully to a narrow
estuary in the metronome and to a meandering river experiment, and indeed the
Weisscher et al reference reports on this, while the method to incorporate
vegetation resistance to flow was already tested in other numerical modelling
(references to van Oorschot and Brückner were added to the paper). The main
reason for using the model is that PIV in the vegetated experiment is
impossible as the plastic floating particles would get trapped within the
vegetation.

Are the sand and mud experiments identical to some of the runs in Braat et al.
(2019) or are they new runs with similar sediment and forcing? The wording was
a little unclear, and I am having some trouble matching up the experimental
conditions between those described here and those in Braat et al. (2019).

REPLY We made clearer in the paper that the experiment with only mud is indeed
that experiment first described in Braat et al. (2019), reanalysed on
different aspects here. We checked the description of the experimental
conditions carefully and made minor corrections.

MINOR typographic corrections were made as indicated.

RC2: Anonymous Referee, https://doi.org/10.5194/esurf-2021-75-RC2
General Comments

In the manuscript "Estuarine morphodynamics and development modified by
floodplain formation", the authors present a combination of physical and
numerical modeling results to demonstrate the role of vegetation and mud in
estuarine evolution. The authors suggest two separate mechanisms that affect
morphology: mud changes estuarine dynamics by filling in accommodation space,
while plants increase roughness. This is interesting because both mud and
vegetation are often thought to stabilize systems and prevent change, but the
authors demonstrate that they do this in different ways. Overall, I think the
paper contributes novel results that directly relate to field observations of
estuarine sediment transport and dynamics, and by using a physical model you
are able to separate confounding variables that are impossible to separate in
field work.

REPLY We thank RC2 for the constructive comments, suggestions and discussion.

Specific Comments; Plants:

One component of the physical experiments that seems perhaps too simplified is
the vegetation and the role of the roots. First, root morphology is important
for stabilization, which is neglected in this paper. For these experiments, it
seems that you chose the two species of plants with the shortest roots and
smallest diameters (based on Lokhorst 2019). Do you think your results would
be different if you had used plants with more extensive (or interlocking) root
structures? How do your plant choice relate to plant root proprieties in the
field?

REPLY Plant roots: we will discuss the effects of rooting in more detail. Our
first, geometric scaling consideration was that roots in natural estuaries
have lengths of a fraction of the main channel depth, and our smallest plants
still have relatively large roots. Our second, dynamic scaling consideration
was that of bank erosion and channel incision reduction. Earlier experiments
with vegetation (e.g. in preparation for van Dijk et al. 2013) showed that
more extensive and interlocking root systems can completely fixate systems. In
view of this experimental difficulty, we chose the smallest plants, which have
measureable bank erosion reduction effects as shown in Lokhorst et al (2019)
in bespoke bank erosion tests at the scale of the experiments. The third, also
dynamic scaling consideration was that of hydraulic resistance. As long as the
stems penetrate the water surface and there is sufficient stem density, the
vegetation has a strong measureable hydraulic resistance effect. Unlike the
roots that are small relative to channel depth in large natural estuaries, the
vegetation settles at such elevations that its effect on the hydrodynamics can
be large in the field, and this is also the case in the experiments.

Second, stabilization by plants can be affected by the sediment type. Previous
studies have demonstrated that in sandy sediments, vegetation may not provide
stabilization (e.g., Feagin et al. 2009). I think your experiments may
underestimate the potential role of plants in bank stabilization given that
you use sandy sediments. Have you considered doing experiments that have both
mud and vegetation? That is likely beyond the scope of this paper, but I would

be interested to see if plants have a more profound effect with finer grained sediments.

REPLY Stabilization by plants: we agree that the stabilization of sediment by plants under waves would depend on sediment type. However, that kind of stabilization is only relevant on cut banks (and note that there were no waves in the inner estuary in our experiments - that is yet another variable to test by ventilator in the future). In rivers and estuaries, another kind of stabilization also matters: that of surface protection against channel carving (Kleinhans et al. 2018). This protection is done basically by increased hydraulic resistance, and this effect could be amplified if the sediment is also cohesive, and cohesive sediment alone can also have this effect. However, given time investment needed for these experiments, we have yet been unable to conduct an experiment with this narrow estuary setup with both mud and plants. This is unfortunately not beyond the scope of the paper, but it is beyond what we could manage.
Both this and the previous point are now explained in a separate paragraph in the methods section after introduction of the vegetation species.

I was also curious about why you used two different plant species and then did not discuss the difference between the two other than in the methods. For roughness, you used stem density and diameter, which would have taken into account some of the differences between the plants. You also state that the plants have different zonation. Was the use of the two species just to get maximum vegetation cover (plants that would inhabit all elevations of bars and floodplains)? Was there any difference in the flexibility or surface area of the species of plants that may alter the roughness?

REPLY Different plant species: Indeed we used multiple species. We can describe the eco-engineering effects of the selected species from Lokhorst et al. in a bit more detail and we will also explain better what the purpose was. As the reviewer already understood, it is mainly to have a larger vegetation cover. Simplistically one could argue that supplying more seeds would also do that but our controlled experiments in Lokhorst et al. showed that they settle in different places. As such, the landscape becomes biogeomorphologically richer and indeed we saw zonation happen. The effect may be limited to this: for the same density we got similar hydraulic resistance from both species in Lokhorst's experiments so we don't expect flexibility and such traits to play a role here. In this sense, the laboratory vegetation simplifies the vegetation in nature, but as this study addresses the scale of entire estuarine systems, such simplifications are as unavoidable as they are in numerical biogeomorphological models doing entire estuaries. Being an experiment rather than a model, the results are complementary to numerical models.

Specific Comments; Other comments:

Did you measure the grain size distribution of the sand at the end of the mud experiments? I am curious if mud infiltrated the porespace of the sand. This would have morphodynamic importance – if the mud simply deposits on the top, it would behave like mud-capped sands, but if the mud infiltrated, the sand may behave cohesively.

REPLY We have looked at the possibility of infiltration, but because the crushed nutshell is rather course compared to the sand, it does not measureably infiltrate, as now mentioned in the methods.

Small question, but could you clarify the lights used in the lab – are they full spectrum? Were the lights on all the time, or was there a dark cycle ("night")? From the text, it seems that the lights were always on, but experiments often use a light/dark cycle to simulate days. I suppose this would be difficult in scaling your experiments, so how do you think having the

light on all the time affected plant growth? (See Smith and Sitt 2007 for some discussion of this).

REPLY We specified the lighting in the methods, which was daylight-toned TL as also used by biologists in our university in their plant growth labs. We checked the Smith and Stitt paper, but the biologists have their lights on at all times (no night) and so did we, and shutting it off weakened the plants in tests as they stretched for light and got weak stems. The aim was to get the plants to grown seedling stage a.s.a.p.

Technical Corrections were implemented as suggested.

---

## Author Response (AR2)

Review by Associate Editor Claire Masteller (minor revisions)

We thank the AE for her time and constructive comments. Below we respond point by point and indicate the changes made in the manuscript. Minor points from the PDF were all answered and used to clarify the text. Maarten Kleinhans, on behalf of all authors

General comments:

- I found the introduction to be a bit muddled, and that the ideas and motivation could be streamlined with a more logical progression. I also found the terms "floodplain formation" and "floodplain" to be used almost interchangeably through this section, which I think could be modified for clarity by focusing on the role of mud or vegetation in driving the formation of the floodplain. I have made suggestions in the attached PDF toward this end.

REPLY: Thank you. We have implemented a number of changes that should clarify the text and the difference between floodplain formation as part of the dynamics and floodplain as having an effect by its presence.

- In the results, I had difficultly wrapping my head around the scale of the differences/changes between different experiments, the stages of each experiment, and different experimental reaches. The authors include figures produced from high-precision, high-quality measurements, and I encourage the authors to use the measurements to add more quantitative results to this section. For example, the terms "reduced", "Small", and "intermediate" do not provide a good sense of how marked the evolution of these features are throughout the experiment or the scale of difference between the experiments. I've made a number of comments in the attached PDF pointing out areas where I think this could be addressed to aid in reader understanding.

REPLY: We have done this more now. In the previous version we sought a balance between quantification of results that is useful for the understanding while not encouraging over-interpretation of our idealized small-scale estuaries.

- Additionally, I encourage the authors to revise the MS such that the experiments are discussed in a consistent order and that that order be consistent with the figures. I found often that the mud experiment was discussed first, then the control, and then the vegetated experiment. However, the figures present the experiments in the order (1) Sand, (2) Vegetation, (3) Mud. In order to ease the parsing of the results by the reader, I encourage the authors to choose an order (probably Control, Vegetation, Mud based on the figures) and stick with this order when introducing the experimental design and set-up, the results, and the discussion.

REPLY: The entire manuscript was checked and adjusted for a consistent order of sand (control), vegetation and mud. Where general development is described in the results and the experiments are compared in dimensions or dynamics, the present order is maintained.

Figure comments

Please note that the colors for mud and vegetation are switched between figure 3 and figure 9. Please make these consistent.

REPLY: thanks for your careful check. Figure 9 has been adjusted.

---

## Author Response (AR3)

Review by Associate Editor Claire Masteller (accept with technical correction)

Regarding your figure #1: please add the copyright icon as follows: © Google Earth.

Done. It was already done in the previous version as part of the last caption sentence. I am not sure why that is not correct so I added this to the end of the caption.